# Snow Water Equivalent Retrieval Over Idaho, Part B: Using L-band UAVSAR Repeat-Pass Interferometry

Zachary Hoppinen[*1,2], Shadi Oveisgharan[3], Hans-Peter Marshall[1], Ross Mower[4,5], Kelly Elder[6], and Carrie Vuyovich[7]

[1]Boise State University, Department of Geosciences, 1295 University Drive, Boise, ID, USA
[2]Cold Regions Research and Engineering Laboratory, Engineer Research and Development Center, United States Army, Hanover, NH 03755, USA
[3]Jet Propulsion Laboratory, California Institute of Technology, 4800 Oak Grove Dr, Pasadena, CA, USA
[4]The National Center for Atmospheric Research, Boulder, Colorado, USA.
[5]Department of Civil and Environmental Engineering, University of Washington, Seattle, WA, USA
[6]Rocky Mountain Research Station, US Forest Service, Fort Collins, CO, USA
[7]Hydrological Sciences Laboratory, NASA Goddard Space Flight Center, Greenbelt, MD, USA

**Correspondence:** Zachary Hoppinen (Zachary.Keskinen@boisestate.edu)

**Abstract.** This study evaluates using interferometry on low frequency synthetic aperture radar (SAR) images to monitor snow water equivalent (SWE) over seasonal and synoptic scales. We retrieved SWE changes from nine pairs of SAR images, mean 8 days temporal baseline, captured by an L-band aerial platform, NASA's UAVSAR, over central Idaho as part of the NASA SnowEx 2020 and 2021 campaigns. The retrieved SWE changes were compared against coincident in situ measurements (SNOTEL and snow pits from the SnowEx field campaign) and to 100 m gridded SnowModel modeled SWE changes. The comparison of in situ to retrieved shows a strong Pearson correlation (R = 0.80) and low RMSE (0.1 m, n = 64) for snow depth change and similar results for SWE change (RMSE = 0.04 m, R = 0.52, n = 57). The comparison between retrieved SWE changes to SnowModel SWE change also showed good correlation (R = 0.60, RMSD = 0.023 m, n = 3.2e6) and especially high correlation for a subset of pixels with no modeled melt and low tree coverage (R = 0.72, RMSD = 0.013 m, n = 6.5e4). Finally, we bin the retrievals for a variety of factors and show decreasing correlation between the modeled and retrieved values for lower elevations, higher incidence angles, higher tree percentages and heights, and greater cumulative melt. This study builds on previous interferometry work by using a full winter season time series of L-band SAR images over a large spatial extent to evaluate the accuracy of SWE change retrievals against both in situ and modeled results and the controlling factors of the retrieval accuracy.

## 1 Introduction

Seasonal snow is a critical resource providing drinking water for millions, clean hydro-electric power generation, and supporting multi-billion dollar agricultural and recreation industries, with the total value of seasonal snow estimated in the trillions of dollars (Li et al. , 2017; Sturm et al. , 2017). Consequently, understanding the distribution of seasonal snow water storage and subsequent runoff is essential.

Current techniques fail to effectively resolve snow properties for water forecasters and managers in the face of a changing global climate that will fundamentally alter previous relationships between snowpack monitoring sites (SNOTEL), point-based automatic weather stations and runoff forecasts. Climate change will bring shifts in the timing and intensity of melt rates (Kunkel , 2016), more frequent rain-on-snow events (Cohen , 2015), and complex region-specific evolution in snowfall patterns (Strapazzon et al. , 2021; Domingues et al. , 2012). These changes are altering the relationship between point observations (e.g. SNOTEL sites) and snow conditions at other elevations, and therefore the statistical techniques for using sparse in-situ snow measurements to predict spring run-off are becoming less accurate (Livneh and Badger , 2020). A shift to spatially distributed snow estimates is required, however due to the short length scale of variability in snow, increasing the in-situ network to the necessary station density is not logistically or economically feasible. Remote sensing and modeling represent a promising method of determining spatially distributed snow estimates to capture the complexities of snow distributions. Remote sensing of snow water storage currently relies primarily on passive-microwave systems with coarse tens of kilometer-scale resolution. Additionally, passive microwave systems can only measure snow depths up to a meter and; therefore, are not valuable for mountainous areas where most of the water available for water resource usage is stored. Active microwave-based synthetic aperture radar (SAR) does not have depth limitations or resolution constraints making it useful for measuring and resolving complex global snow water storage patterns.

## 1.1 SAR overview

SAR sensors actively emit electromagnetic energy in the microwave range, frequencies from 1 to 40 GHz, and measure the backscattered (returning) energy and phase. For lower frequency, longer wavelength microwave SAR systems over snow-covered ground, the electromagnetic waves are refracted across the air-snow interface and then reflected from the snow-ground interface with limited snow grain interactions (Naderpour et al., 2022). Passing through the snowpack leads to two competing effects: shorter two-way travel distance but slower wave speeds. These effects combine to create phase shifts, due to the changes in travel time, between repeat SAR images that we can analyze to quantify snow height and snow water equivalent changes (Figure 1). Analyzing phase changes between SAR images, called interferometric SAR (InSAR), allows us to retrieve changes in the snowpack volume using these SAR-observed snow height changes and modeled or observed densities.

Since the returned phases are measurements of sinusoidal wave offsets there is a "wrapping" effect where as the phase approaches and then passes $2\pi$ it returns to zero again. This causes an $2\pi$ modulo ambiguity that has be resolved by unwrapping the phase and adding or subtracting increments of $2\pi$ to the phase to recover the absolute or unwrapped phase (Goldstein et al., 1988; Rosen et al., 2000). This unwrapping process is relatively simple in high coherence regions but may be impossible in lower coherence regions. Consequently, InSAR analysis may use either the wrapped phase, when continuous data is required and the expected phase change is less than $2\pi$, or unwrapped data when the expected phase change may exceed $2\pi$.

In this study, we use Equation 1, proposed by Guneriussen et al. (2001), to retrieve snow height changes ($\Delta d$) at a specific wavelength ($\lambda$) from incidence angle ($\alpha$), phase change ($\Delta\phi$), and the real component of the dielectric permittivity ($\epsilon_s$). $\epsilon_s$ measures the wave speed through the snowpack and depends on only the snowpack density and liquid water content. We

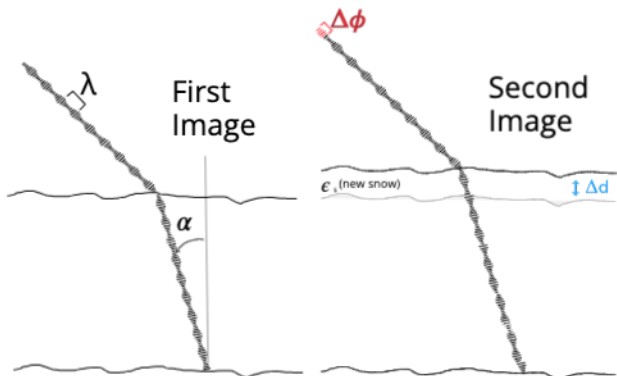

**Figure 1.** Conceptual model of phase differences between two SAR images with an increase in snow height between image acquisitions. $\Delta d$ is the increase in snow height between the first and second images, $\Delta\phi$ is the corresponding shift in phase due to changes in path length and wave speed, $\alpha$ is the incidence angle, $\lambda$ is the wavelength, and $\epsilon_s$ is the real-component of the dielectric constant of the new snow.

estimated the real component of the dielectric permittivity from snowpack density ($\rho_s$) in kg m$^{-3}$ using Equation 2, assuming no liquid water in the snowpack (Matzler , 1996).

$$\Delta d = -\frac{\Delta\phi\lambda}{4\pi}\frac{1}{\cos\alpha - \sqrt{\epsilon_s - \sin^2\alpha}} \tag{1}$$

$$\epsilon_s = 1 + 1.6\times 10^{-3} * \rho_s + 1.8\times 10^{-9} * \rho_s^3 \tag{2}$$

InSAR retrievals of the amount of water stored in the snowpack, snow water equivalent (SWE), are generally preferred to capturing snow height changes for two reasons. First, most water managers are primarily interested in the volume of water stored within the snowpack rather than the height of snow. Secondly, conversion to SWE should minimize the effects of errors in the estimation of $\rho_s$ in Equation 2. For example, an overestimation of $\rho_s$ will lead to a slower estimated wave speed through the snowpack and an underestimation of snow depth. However, when we convert the retrieved snow depth to SWE using this higher density, this underestimation of depth is counteracted, minimizing the impact of errors in density. This means using even the edge cases of snow density have a limited impact on SWE retrievals ($< 7$ %; Leinss et al., 2015). Some studies have used approximations of SWE change directly from phase (Leinss et al., 2015; Guneriussen et al., 2001; Oveisgharan et al., 2023). Since we wanted to compare to our snow depth measurements and errors in density should have limited impacts on the retrievals we use a density estimate to calculate both snow depth and SWE changes (see Section 3.4).

## 1.2 Previous work

Previous research has demonstrated promising retrievals of snow height and SWE changes using InSAR phase shifts in seasonal mountain snowpacks. However, these studies have focused on simple topography with limited numbers of InSAR pairs and

70 have yet to explore the accuracy or controlling factors of InSAR retrievals in complex mountain terrain (Marshall et al. , 2021; Deeb et al. , 2011).

Guneriussen et al. (2001) showed fringe patterns consistent with snow accumulation in a pair of March ERS-1, C-band ($\lambda \approx$ 0.05 m), images that captured a 4 centimeters change in SWE but did not have in situ observations to validate the change. Marshall et al. (2021) explored L-band, ($\lambda \approx 0.231$ m), snow depth inversion over a 4-km$^2$ region on Grand Mesa, Colorado,

using lidar and UAVSAR imagery from February 1 and February 12, 2020, and showed good agreement between lidar depth change and UAVSAR estimated depth change ($r^2$=0.76, RMSE < 0.05 m). Tarricone et al. (2022) used three L-band InSAR pairs to estimate both accumulation and ablation in the Jemez River, New Mexico, with accumulation and ablation patterns showing agreement with both in-situ depth sensors and changes in fractional snow covered area. Ruiz et al. (2021) successfully captured a season-long SWE accumulation at a tower-based site using InSAR phase changes at L-band with temporal baselines

up to 12 days in length at L-band. Comparison to in situ SWE showed RMSEs between 8.77-26.07 mm depending on the temporal baseline used in the analysis. Additionally, they showed an overestimation of SWE by 7-21 mm of SWE from the tower-based L-band InSAR compared to the in situ SWE measurements. Finally, Nagler et al. (2022) measured SWE changes using a C and L-band airborne InSAR instrument for a pair of snowfall events. The in situ measurements captured 66.4 mm of SWE change, and the L-band InSAR captured a mean SWE change of 70.3 mm with a root mean squared difference of

11.2 mm. These studies demonstrate the promise of InSAR SWE and depth retrievals at both C and L-band. We expand on these previous studies by increasing the number of in situ observations and image pairs analyzed and utilizing the large study areas which include complex topography and a large elevation range to explore controlling factors on the accuracy of InSAR retrievals.

### 1.3 Research questions

We explore two research questions to clarify the ability of lower frequency, L-band radars to retrieve snow properties:

1. How accurate are snow depth and SWE change retrievals over complex mountain terrain using L-band interferometric radar analysis?

2. How does tree coverage, total snow depth, incidence angle, coherence, and snow wetness impact the accuracy of L-band interferometric retrievals?

## 2 Methods

### 2.1 Datasets

This research combines in situ, modeled, and aerial data sets of snow properties over the central mountains of Idaho to address our research questions (Figure 2) (Liston and Elder , 2006; Liston et al. , 2020). We use the weekly to biweekly time series of L-band SAR observations from UAVSAR collected for the 2020 and 2021 NASA SnowEx Mission, to retrieve snow depth

and SWE. The validation data included in situ observation from weekly snow pits (depth, SWE, wetness), telemetered snow

station measurements, interval boards (Greene et al. , 2022) and spatially distributed model results of both SWE and SWE melt. Additionally, we explored how the retrieval accuracy changed with vegetation and topography, using geomorphological and vegetation data from the 10-m USGS National Elevation Dataset (NED) (Gesch et al. , 2018), 2016 National Land Cover Database 30-m vegetation percentage maps (Jin et al. , 2019) and 2019 30-m Global Land Analysis and Discovery forest height dataset (Potapov et al. , 2021).

## 2.2 UAVSAR imagery

The UAVSAR platform is a fully polarimetric L-band InSAR instrument mounted on a NASA Gulf Stream III aircraft (Hensley , 2008; Rosen et al. , 2006). It flies at $\approx$ 13,700 m with a sensor center-frequency of 1.26 GHz ($\lambda = 0.2384$ m). UAVSAR was tasked to perform weekly to bi-weekly observations over 14 sites in the Western U.S. from January to March for the 2020 and 2021 Nasa SnowEx Mission. The UAVSAR imagery was processed by the NASA Jet Propulsion Laboratory, including SAR focusing and georeferencing with the Shuttle Radar Topography Mission (SRTM) DEM, radiometric calibration, motion compensation, and phase unwrapping using the Integrated Correlation and Unwrapping (ICU) algorithm (Rosen et al. , 2006; Fore et al. , 2015). InSAR phase between images was calculated by multiplying the phase of one image against the complex conjugate of another image. The imagery was downloaded and converted to netCDFs using *uavsar_pytools* (Hoppinen et al. , 2022).

This study used 12 UAVSAR image pairs from the 2019-2020 and 2020-2021 winters over central Idaho from repeated flight paths at a heading of 232° (Table 1). These image pairs were all nearest temporal neighbors from 14 image acquisitions. For most analyses we only utilized the nine image pairs that successfully unwrapped (see section 1.1 for discussion of phase wrapping) (Table 1). We utilized the wrapped images when complete spatial or temporal coverage was necessary and explicitly state in the methods when wrapped imagery was used. The bounding box for these images was $\approx$170 x 16 km and stretched from the foothills of Boise into the Sawtooth Mountains. This box includes Dry Creek Experimental Watershed, Mores Creek Summit, and Banner Summit, where field teams performed in situ observations on the flight dates (Figure 2).

## 2.3 In Situ Observations

The in situ data included automated weather station data from SNOTEL stations, SWE increase measurements from interval boards, and snow pits coincident with UAVSAR flights from the Dry Creek, Mores Creek, and Banner Creek study sites (Schaefer and Paetzold, 2000). Interval board data collected the accumulated snow depth and SWE between UAVSAR flights from 4-9 locations near the Banner summit SNOTEL (Figure 3). The SNOTEL stations used were SNTL:ID:978 (Bogus), SNTL:ID:637 (Mores), and SNTL:ID:312 (Banner) (USDS National Resource Conversation Service , 2022). The SNOTEL network is a system of 900 telemetered stations with snow depth, snow water equivalent, and temperature. Stations measures SWE with a pressure measurement from a glycol filled bladder, measuring the weight of the snowpack at hourly intervals. The snow pit observations included measurements of total snow depth, dielectric permittivity, full density profiles in 10 cm increments, layer-based measurements of snow wetness and grain form, and site descriptions of vegetation and ground cover (Table 2). For each InSAR image pair, repeat snow pits within +/- 2 days of both flights were included (Figure 4).

**Table 1.** UAVSAR Flight Dates

| Flight 1 Date | Flight 2 Date | Unwrapped? |
|---------------|---------------|------------|
| 2019-12-20 | 2020-01-31 | N |
| 2020-01-31 | 2020-02-13 | N |
| 2020-02-13 | 2020-02-21 | Y |
| 2020-02-21 | 2020-03-11 | Y |
| 2021-01-15 | 2021-01-20 | Y |
| 2021-01-20 | 2021-01-27 | Y |
| 2021-01-27 | 2021-02-03 | Y |
| 2021-02-03 | 2021-02-10 | Y |
| 2021-02-10 | 2021-03-03 | N |
| 2021-03-03 | 2021-03-10 | Y |
| 2021-03-10 | 2021-03-16 | Y |
| 2021-03-16 | 2021-03-22 | Y |

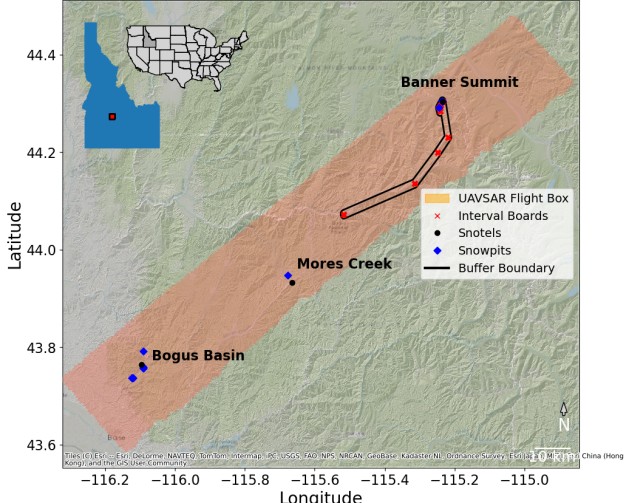

**Figure 2.** Map of the study area showing the outline of the UAVSAR flight box, snow pits, SNOTEL, interval board locations and the buffer around the interval boards used.

## 2.4 SnowModel

A distributed snow-evolution modeling system (SnowModel) was used to simulate snow properties (e.g., snow depth, SWE, snow melt, snow density, etc.) over different climates and landscapes (Liston and Elder , 2006; Liston et al. , 2020). The SnowModel domain is on a structured grid with spatial resolutions ranging from 1 to 200 meters (although it has the ability to

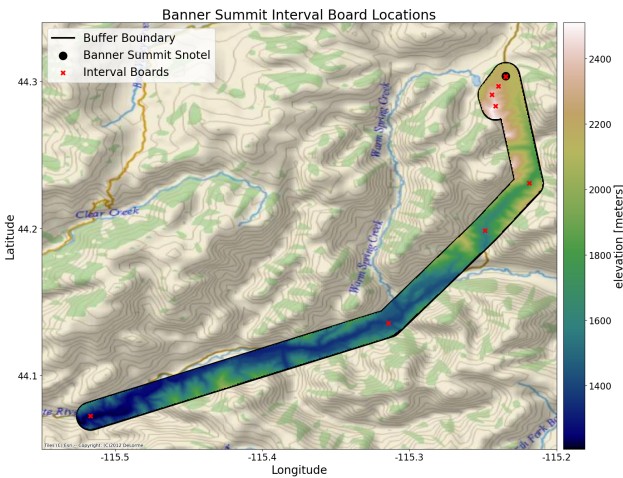

**Figure 3.** Map of the interval board locations and one-kilometer buffer around a line connecting the interval board locations used to clip the UAVSAR data.

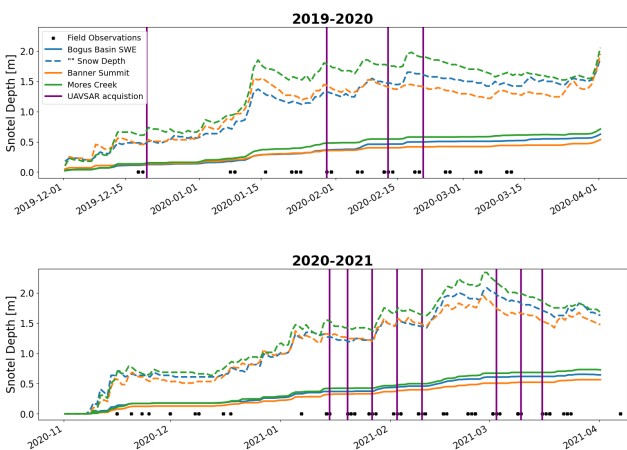

**Figure 4.** 2020 and 2021 snow water equivalent (solid line) and snow depth (dashed line) measurements for the three SNOTELs used in this study. Purple lines represent UAVSAR flights and black xs represent days with field observations.

simulate coarser resolutions, as well) and temporal resolutions ranging from 10 minutes to 1 day. The required inputs to run SnowModel include 1) temporally varying meteorological variables of precipitation, wind speed and direction, air temperature, and relative humidity taken from meteorological stations or atmospheric models and 2) spatially distributed topography and land-cover type. The primary modeled processes include accumulation from frozen precipitation; blowing-snow redistribution and sublimation; interception, unloading, and sublimation within forest canopies; snow-density and grain-size evolution; and snowpack ripening and melt (Liston and Elder , 2006).

**Table 2.** SnowEx snow pits

| Site Name | Date Range | Number of Coincident Pairs | Latitude | Longitude |
|---|---|---|---|---|
| Banner Summit SNOTEL | 2020-01-30 - 2021-03-22 | 12 | 44.29086 | -115.24387 |
| Lower Deer Point - Open | 2021-01-21 - 2021-03-23 | 8 | 43.73691 | -116.12208 |
| Lower Deer Point - Tree | 2021-01-21 - 2021-03-23 | 6 | 43.73634 | -116.12072 |
| Bogus Basin Lower Trees | 2021-01-22 - 2021-03-18 | 5 | 43.75689 | -116.09078 |
| Bogus Basin Lower | 2021-01-22 - 2021-03-24 | 8 | 43.75705 | -116.09099 |
| Mores Creek Summit | 2020-02-12 - 2021-03-04 | 2 | 43.94735 | -115.67666 |
| LDP Open | 2020-01-31 - 2020-03-11 | 4 | 43.73701 | -116.12188 |
| LDP Tree | 2020-01-31 - 2020-03-11 | 4 | 43.73640 | -116.12050 |
| Banner Summit Open | 2020-01-30 - 2021-03-18 | 5 | 44.30461 | -115.23598 |
| Bogus Basin Upper | 2020-01-31 - 2020-03-11 | 4 | 43.75878 | -116.09017 |

The modeled daily aggregated SWE and melt data were generated by simulations using Parallel SnowModel, a parallelized
version of SnowModel (Mower et al. , 2023). The data was indexed from Parallel SnowModel simulations executed using
1,800 processes on NASA's Center for Climate Simulation (NCCS) Discover supercomputer with a 1,560-teraflop SuperMicro
Cluster feature 20,800 Intel Xeon Skylake processes (Carriere , 2023) over the contiguous United States (CONUS) at a 100-
meter grid increment with a 3-hour forcing time step, daily aggregated output, and single-layer snowpack configuration. The
following inputs were used for the simulations: USGS NED for topography on a 30-meter grid (Gesch et al. , 2018), the North
American Land Change Monitoring System (NALCMS) Land Cover 2015 map for vegetation on a 30-meter grid (Homer et al.
, 2015; Jin et al. , 2019; Latifovic et al. , 2012), and forcing variables from a high-resolution Weather Research Forecast(WRF)
model from the National Center for Atmospheric Research (NCAR) on approximately a 4-kilometer grid (Rasmussen et al. ,
2023).

## 3  InSAR snow depth retrievals

### 3.1  Polarization

For this analysis, we utilized the phase changes from the VV polarization, based on higher coherence in the co-polarized band than in the cross-polarized bands. The choice of VV over HH was due to higher coherence in VV and to minimize interactions of the radar echos with vegetation and ice layers, which might raise the phase centroid above the ground surface.

### 3.2  Setting the reference UAVSAR phase

An essential consideration of InSAR imagery is the need for a reference phase. The initially measured phases are arbitrary and only relative differences across a scene have information. Consequently we need a method of setting the absolute phase values of a scene. For snow property retrievals this involves using external information (snow depth or SWE changes) to set the mean phase (Leinss et al., 2015). We set each InSAR image's the mean scene wide phase using the mean snow depth change and densities of all available situ measurements (Equation 3).

$$\phi_{scene}(t) = \frac{\triangle d_{insitu}(t) \times \lambda}{4\pi} \times \frac{1}{\cos\alpha - \sqrt{\epsilon_s(\rho_s) - sin^2\alpha}} \tag{3}$$

with $\triangle d$ representing the average change in snow depth across the in situ stations, $\epsilon$ calculated from the average density across the in situ measurements using Equation 2, and $\alpha$ representing the scene-wide mean incidence angle. We chose to aggregate our insitu and modeled values to set the mean phase of the entire scene rather than tying to specific sites. This approach limits the biasing effects of setting this mean phase with our in situ data since this is only a mean shift over the entire scene. We calculated the mean phase from regions with non-zero modeled SWE to minimize impacts from non-snow related factors.

For the model comparison, we mean shifted the phase to match the expected phase from scene-wide mean in modeled SWE change.

### 3.3  Atmospheric correction

We corrected for atmospheric phase delays due to temporally varying water vapor, air temperatures and pressures by downloading ERA5 atmospheric data for all acquisition dates and calculating the z-integrated phase delay difference between acquisition dates for each pair of images (Hoppinen et al. , 2022; Hersbach et al. , 2020; Doin et al. , 2009). This phase delay difference was then subtracted from the measured InSAR phase changes.

We elected to use the ERA5 reanalysis atmospheric models to remove atmospheric phase instead of phase elevation relationships to avoid removing our SWE accumulation signal which can be correlated with elevation. To visualize the impacts of these atmospheric corrections, we converted these atmospheric corrections to theoretical errors in the SWE retrieval using the incidence angle raster and the mean in situ density for each pair of images (Figure 5).

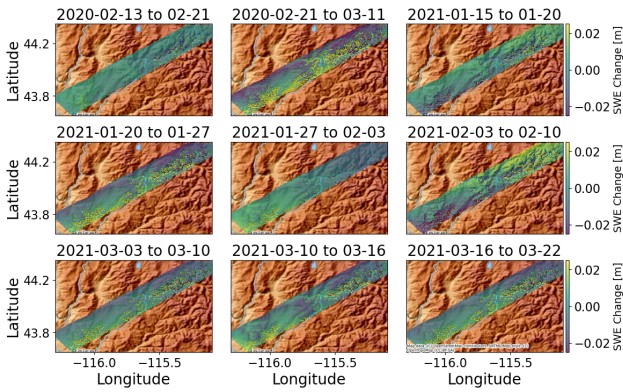

**Figure 5.** Changes in the retrieved SWE from each time period from atmospheric corrections. Note that each atmospheric correction was normalized to a mean of zero and plotted with the same color bounds to improve comparisons.

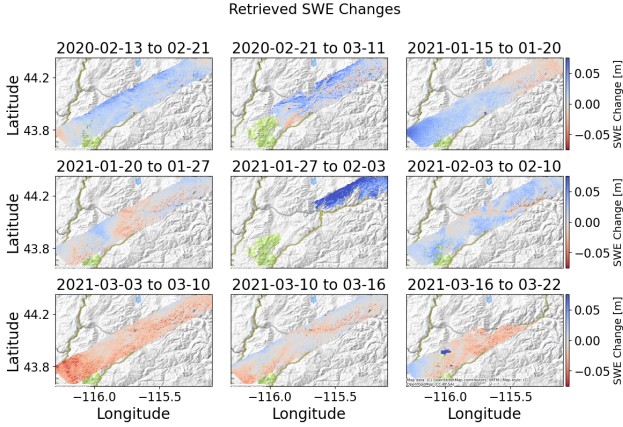

**Figure 6.** The retrieved SWE change in meters for each pair of nearest neighbor image pairs. To improve comparison these are all plotted on the same color bounds.

### 3.4 Retrieving SWE changes

For each UAVSAR image pair, we used the mean in situ density from the relevant SNOTEL and snow pit observations to estimate the dielectric permittivity combined with the local incidence angle and phase to calculate the snow depth and SWE change at each pixel using Eq 1 (Figure 6).

### 3.5 Statistical comparison of InSAR SWE retrievals

We began by visualizing three SNOTEL sites' SWE and snow depth measurements for January 2021 to March 2021 against co-located UAVSAR snow and SWE retrieved depths. To generate our UAVSAR retrieved snow depth profiles, we used the

in situ measured depth and SWE at our first flight date (January 15th, 2021) and then cumulatively added the mean retrieved snow depth and SWE from a 100 m box around each SNOTEL site ($\approx$ 20 looks). As some image pairs and regions failed to unwrap (Table 1, Figure 6), we used the wrapped phase to ensure we would be able to retrieve a continuous SWE retrieval time series for all the SNOTEL sites.

We next compared the UAVSAR snow depth and SWE change retrievals to the field snow pits and SNOTELs combined using the Pearson correlation coefficient (r), root mean squared error (RMSE), and mean absolute bias (retrieved - in situ). The retrieved values were averaged from a 100m box around the field location. The snow pits were selected if they were within +/-2 days of either the first or second flight in a pair. We subtracted the snow depth from spatially coincident pits to get the snow depth change observed between flights. We calculated 95% confidence intervals for RMSE, Pearson-r, and bias from one thousand bootstrapped samples of the UAVSAR and in situ snow depth changes (n = 64).

To explore how effectively the UAVSAR-retrieved SWE captured orographic trends in accumulation, we compared the relative increase in SWE from a series of interval boards at varying elevations near the Banner Summit study site to the relative increases in retrieved UAVSAR SWE changes. We set the SWE change to zero at the lowest board and calculated the changes in SWE change at each elevation we had an interval board. We did the same for the UAVSAR retrieved SWE change within a 1-kilometer buffer around a line connecting the interval board locations. The UAVSAR SWE change in that buffer was then binned by elevation and plotted against the increase in SWE on the interval boards. We used the wrapped phase in this analysis to avoid bias due to aspects or elevations with lower coherence.

We compared the SnowModel and UAVSAR retrieved SWE changes by calculating Pearson-r and root mean squared difference (RMSD) for all pixels, a subset of pixels that had no modeled SWE melt, and another subset of pixels with no modeled melt and with tree percentages below 10%. We chose RMSD over RMSE for the comparison of modeled to retrieved SWE due to the error between the SnowModel and in situ measurement (RMSE = 0.15 m, R = 0.45). Using RMSD better captures that differences between retrieved and modeled results do not necessarily represent primarily errors in the retrieved values but represent some combined and unknown contributions of errors from both datasets. Next, to explore how RMSD changed with varying geophysical and snow properties, we compared the changes in RMSD by plotting the temporal RMSD at each pixel against the temporally averaged coherence, tree percentage, elevation, maximum modeled SWE depths, and average SWE melt. We used the wrapped phases in this comparison step to avoid biasing the results from low-coherence areas that failed to unwrap and since the magnitude of SWE change in our study region rarely exceeds $2\pi$. We then binned the RMSDs across the whole parameter ranges for elevation, incidence angle, tree percentage, tree height, coherence, and cumulative melt. We excluded bins with less than 100 pixels to avoid bias from small sample sizes.

## 4 Sensitivity analysis

To explore how phase changes relates to snow properties at L-band, we calculated the theoretical phase change across physically reasonable ranges of snow depth change (0 - 0.5 m), new snow density (100 -300 kg m$^{-3}$), and incidence angles (30-60°) (Figure

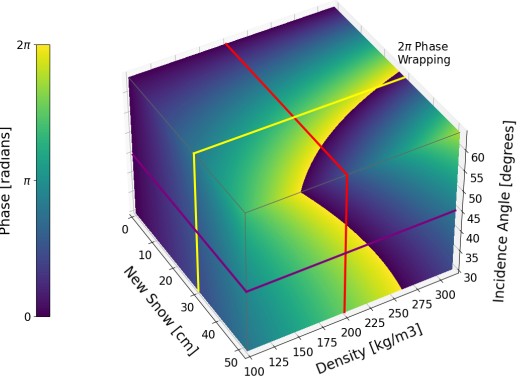

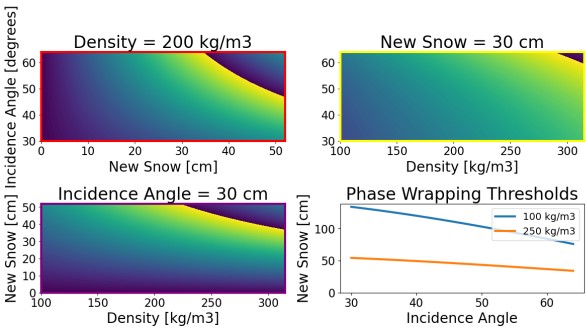

**Figure 7.** Theoretical phase increases for varying new snow amounts, densities, and incidence angles. The $2\pi$ wrapping point data is annotated. Slices through the cube are shown for a constant 200 kg/m$^3$ density (red, middle left), a constant 30 centimeters of new snow (yellow, middle right), a constant 30 degree incidence angle (purple, bottom left), and a display of phase wrapping thresholds for varying incidence angles and new snow heights at 100 and 250 $kg\ m^{-3}$ densities (bottom right).

7). These results show that phase wrapping should not occur until quite significant snow accumulations of 127 mm of SWE at $30°$ and 80 mm at $60°$.

    Liquid water in the snowpack impacts radar's wave speed and phase change and consequently causes errors in retrievals since we only parameterize the dielectric permittivity based on density. We parameterized the dielectric constant for dry and wet snow to evaluate the effect of liquid water. We evaluated the increase in retrieved snow depths with no actual increase for varying degrees of liquid water percentage (Tiuri et al. , 1984). The impacts of liquid water in the snowpack are significant. They can cause up to an $\approx 0.1$ m overestimation of new snow accumulation in the retrievals, with only 0.1 meters of snow becoming wet (Figure 8).

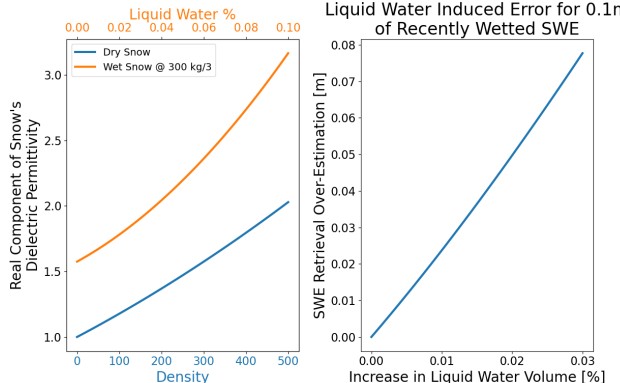

**Figure 8.** On the left, the theoretical dielectric permittivity for varying snow density (blue) and wetness (orange, constant 300 kg m$^{-3}$ density) using the equations from Tiuri et al. (1984). On the right, SWE retrieval errors caused by 0.1 meters of SWE that increased by varying liquid water percentages in wetness between SAR images.

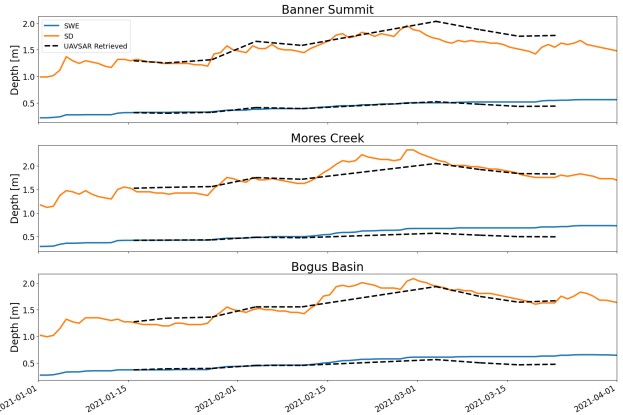

**Figure 9.** Snotel in situ measured snow depth profile plotted against the cumulative retrieved UAVSAR snow depth changes for a 100 m box around each SNOTEL location. Note that the initial snow depth was set to the in situ measured snow depth at the first flight date in 2021.

## 5 Results

We compare UAVSAR retrieved SWE and snow depths against SNOTEL SWE and snow depth changes, SWE and snow depth changes from consistently located snow pits, interval board SWE increases, and SnowModel's SWE change.

### 5.1 SNOTEL visualization

The time series of snow depth from our three SNOTEL sites matches well against our retrieved UAVSAR time series. Since we had longer temporal separations between our UAVSAR observations, some short temporal scale patterns were missed. Overall, the trends and magnitudes of snow depth accumulation are well captured (Figure 9).

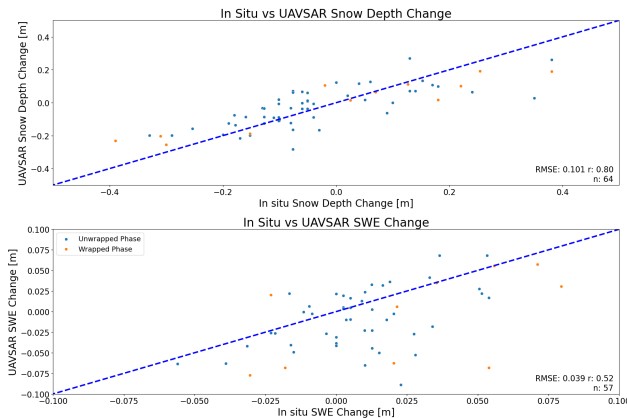

**Figure 10.** Comparison of the UAVSAR snow depth retrievals to in situ snow depth and SWE change measurements. In situ observations without successful phase unwrapping used the wrapped phase in orange. Note that seven SWE change observations were removed for have measured in situ new snow densities above 500 kg m$^{-3}$.

## 5.2 In situ comparison

An expanded comparison, including snow pit and SNOTEL data, shows a positive agreement between retrieved and in situ snow depth observations with an RMSE of 0.1 m and a Pearson correlation of 0.80 (n = 64)(Figure 10). The bootstrapped
analysis had a 95% confidence interval for the RMSE of 0.09 - 0.11 m, with a mean RMSE of 0.10 m, Pearson correlation of 0.71 to 0.87, and bias confidence interval of -0.028 to 0.012 m. The full SWE retrieval were more scattered with an RMSE of 0.041 m and a Pearson correlation of 0.4 (n = 64). Removing in situ retrievals with unreasonably high ($>$ 500 kg m$^{-3}$) inferred density from the in situ measured SWE and snow depth change improves the RMSE to 0.039 and the Pearson correlation to 0.52 (n = 57).

## 5.3 Comparison to interval boards

The UAVSAR retrieved SWE changes captured the overall trends in precipitation relative to elevation for most image pairs, with the atmospherically corrected trends in SWE change generally improving the relationship to the measured orographic trends (Figure 11).

## 5.4 Comparison to SnowModel

We first qualitatively compared the SnowModel SWE changes to those captured in the phase images for two periods of increasing SWE with successful phase unwrapping near the Banner Summit and the Mores Creek SNOTELs (Figure 12). Both examples showed similar accumulation patterns to SnowModel, but consistently more significant SWE changes in the UAVSAR retrieved SWE changes than SnowModel.

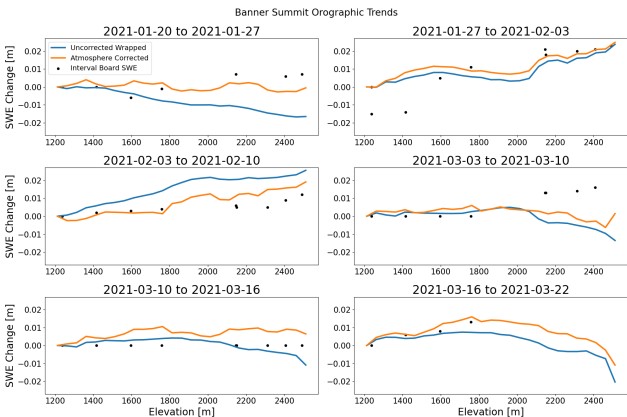

**Figure 11.** Comparison between the retrieved UAVSAR SWE changes with elevation relative to the interval board data sets on SWE changes with elevation. Note the consistent y-limits for all subplots.

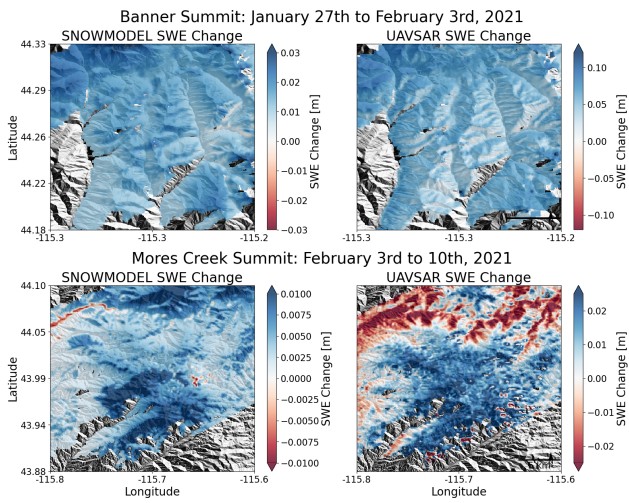

**Figure 12.** Comparison of the UAVSAR and SnowModel SWE changes for two accumulation periods near the Banner Summit and Mores Creek SNOTELs. Note the different visualization ranges for the SnowModel and UAVSAR SWE changes.

The relationship between the SnowModel and UAVSAR SWE change for all the successfully unwrapped pixels showed large regions of highly negative SWE changes in the retrieved data. However, subsetting to only pixels with no SWE melt in SnowModel results removed these regions, suggesting that large areas of wet snow were causing bias or low coherence in the SWE change retrievals (Figure 13).

The spatial maps of RMSD between the SnowModel and retrieved SWE changes show much higher RMSDs at lower elevations along the valley bottoms, in regions to the southeast with higher average SWE melt, and along higher elevations with higher maximum SWE values (Figure 14).

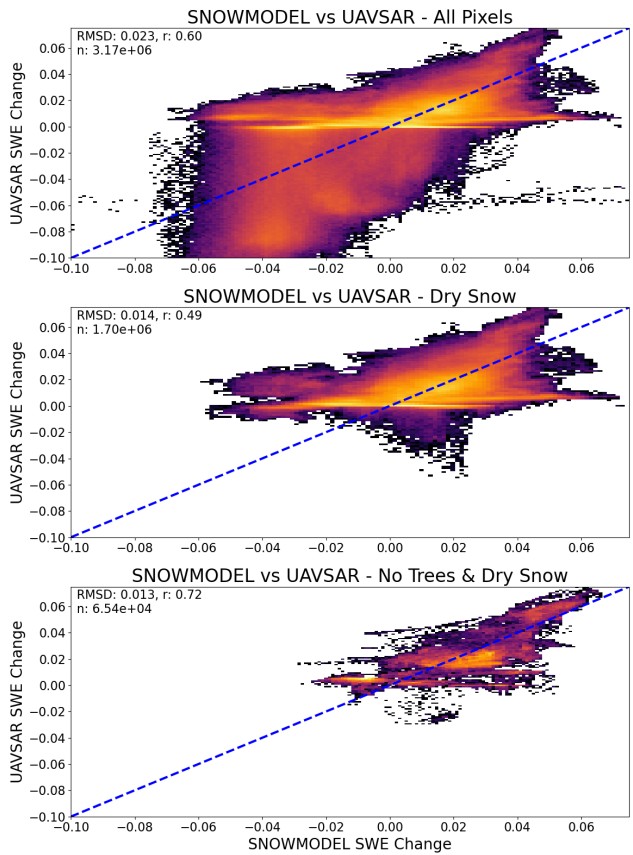

**Figure 13.** 2D log-scaled heat maps showing UAVSAR vs SnowModel SWE changes for all possible unwrapped pixels (top), subset to only pixels with no modeled melt (middle), and subset to pixels with no modeled melt in regions with less than 20% tree percentage (bottom).

The binned analysis of RMSD generally agreed with the spatial comparison with improving RMSD for higher elevations, steeper incidence angles, lower tree heights and coverage percentage, higher coherences, and lower cumulative melt quantities (Figure 15).

## 6   Discussion

### 6.1   How accurate are L-band SWE and snow depth change retrievals over complex mountain terrain?

Our results suggest that L-band InSAR is a promising technique for retrieving snow depth changes over complex mountainous terrain. The comparison between in situ snow depth changes compared to retrieved snow depths show a strong correlation, r = 0.80, and reasonable errors, RMSE = 0.10 m. The comparison to the modeled SWE changes also showed good correlation and low RMSD (RMSD = 0.041 m, r = 0.49). This comparison was even more favorable in dry snow with low tree coverage

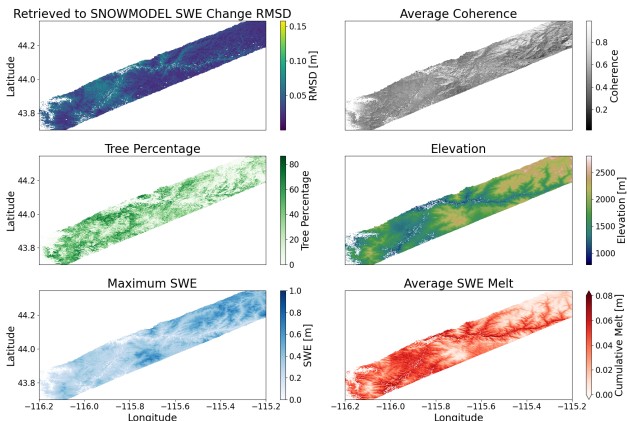

**Figure 14.** Spatial distribution of the RMSD between the SnowModel and wrapped UAVSAR SWE changes, coherence averaged across all time periods, NLCD tree cover percentage, SRTM elevation, the modeled maximum SWE depth, and average SWE melt across all time periods.

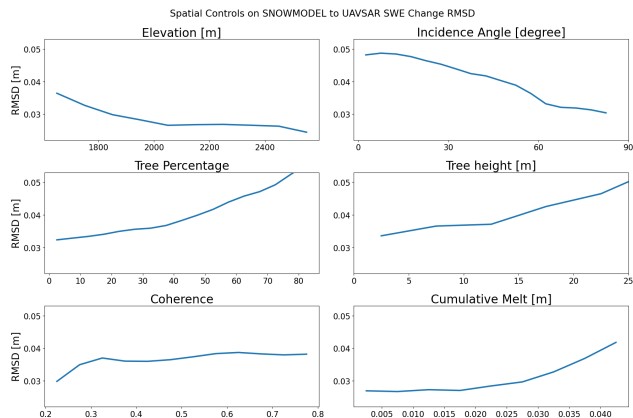

**Figure 15.** Binned distribution of the RMSD between the SnowModel and wrapped UAVSAR SWE changes, coherence averaged across all time periods, NLCD tree cover percentage, SRTM elevation, the modeled maximum SWE depth, and average SWE melt across all time periods.

(RMSD = 0.014 m, r = 0.72). These results suggest that L-band InSAR snow depth change retrievals capture reasonable trends and may be a valuable tool for measuring SWE and snow depth changes.

It is interesting that the retrieved SWE changes compared worse (r = 0.40) relative to the snow depth change (r = 0.80) for the in situ results. This may have been the result of errors in field measurements due to the challenges of measuring SWE changes in the field relative to measuring snow depth changes. In fact a number of observations had unreasonably high new snow densities ($> 500$ kg m$^{-3}$) suggesting either a large rain event or measurement error occurred.

## 6.2 How do vegetation, incidence angle, slope, and snow-wetness impact the accuracy of L-band retrievals?

The predominant factors in the accuracy of this retrieval technique are the need for dry snow and avoiding high vegetation coverage. The highest RMSDs relative to the modeled data occurred in pixels with high tree cover percentages and large amounts of modeled SWE melt, suggesting that wet snow and vegetation may be negatively effecting retrievals.

The negative impacts on accuracy due to liquid water match our theoretical expectations. Since we parameterize the real component of the snow's dielectric permittivity based solely on density, we expect retrievals of snow depth changes in newly wet snow to diverge from in situ and modeled measurements. This divergence will impact not only the measurements at that specific location but also throw off the mean phase change of the image leading to systemic biases across the image when setting the reference phase. Additionally liquid water causes significant drops in coherence which may also have negatively impacted our retrievals (Ruiz et al., 2022). Future researchers could improve retrievals by using modeled melt values to improve the parameterization of potentially wet pixels.

The impact of vegetation also matches the theoretical expectation that the increase in scatterers in the vegetation, primarily branches and trunks at L-band, would cause a decrease in the coherence of those pixels due to more random movements of the scatterers within the pixel and a shifting of the phase centroid up towards the above-snow scatterers, breaking our theoretical requirement of a phase centroid at or close to the snow-ground interface. Interestingly Figure 15 showed a minimal relationship between coherence and RMSD above 0.35 m, suggesting that the decrease in coherence might have been a smaller factor.

Alternatively, the model performs worse in the trees, meaning that the increase in RMSE is due to errors in the modeled data rather than UAVSAR retrievals. The lack of a clear relationship in the in situ data between the vegetation classification at the snow pit and retrieval accuracy also suggests that vegetation impacts may be less significant.

The elevation and incidence angle were other factors related to the retrieval accuracy. The impact of elevation is most likely due to its covariance with liquid water in the snowpack and tree coverage in the scene. Incidence angle was negatively correlated with RMSE. Referring to Figure 7, we see that for a constant change in SWE, the induced phase shift is less at lower incidence angles. This decreased phase signal is closer to the instrument noise floor, and we may struggle to capture these small snow depth changes.

## 6.3 Limitations

This study has a few limitations that should be addressed: the accuracy of SnowModel for comparison, using density in our retrievals of snow depth and setting the reference phase with the in situ data used for validation.

The SnowModel results used in this analysis showed a reasonable correlation to in situ results (r = 0.45, RMSE: 0.15 m), but errors that we attribute to the InSAR results in Figure 15 (in highly vegetated or wet snow regions) are actually some unknown combination of modeling and InSAR retrieval errors compounding. Future work using either more validated SnowModel results or lidar snow depth retrievals will be necessary and our study's modeling comparison should be interpreted with caution focusing mostly on identifying larger spatial and temporal trends in accumulation.

We chose to use aggregated measurements of in situ snow density to directly invert Equation 1 for snow depth and SWE changes. We chose to do this to allow us to compare to our in situ snow depth measurements and due to the low sensitivity to
310 errors in our density approximation discussed in Section 1.1. However unlike studies that use SWE approximations this means that our methods require some approximation of density form either in situ or modeled results to replicate in new study areas (Guneriussen et al., 2001; Oveisgharan et al., 2023).

Another consideration is that we set the reference phase using the average snow depth change and density from across our in situ observations and the average phase of snow-covered regions for each InSAR pair (see Section 3.2) and then used the
315 same in situ observations for validation. Since we are using the reference phase from the whole image and aggregating snow change across multiple sites there shouldn't be too much biasing of our results but it is important to know that the calibration and validation data were the same data. We chose to set the reference phase this way due to the limited number of repeat observations (n = 64) available to use and to avoid biasing a whole image from errors in a single in situ measurement, but future work should explore the impacts and methods of setting the reference phase.

**6.4  Future work**

Future research and practical applications of InSAR-based SWE retrievals should consider: 1. the importance of atmospheric corrections, 2. how to integrate this technique into pre-existing hydrologic monitoring efforts, 3. effective use of in situ data for processing and validating these InSAR retrievals, 4. image masking in wet and low coherence regions, 5. validating these findings with different snow climates and vegetation profiles, 6. the use of wrapped vs unwrapped phase, 7. parameterizations
for the dielectric permittivity in wet snow, and 8. how best to incorporate other frequencies and SAR-based analysis techniques.

Atmospheric correction of InSAR data to account for temporal and spatial variations in radar wave speeds through the atmosphere due to pressure, temperature, and moisture changes is critical for accurately capturing SWE changes. Due to the relationship between elevation and path length in airborne and spaceborne InSAR systems, the atmospheric phase changes will often have similar patterns to orographic precipitation. This relationship makes the removal of the atmospheric phase critical
to accurately capturing SWE change patterns over large areas, especially the accumulation gradient with elevation. The fact that SWE change correlates with elevation also means that practitioners should avoid atmospheric corrections that rely on removing phase vs. elevation relationships, which is common in other InSAR applications. Atmospheric corrections should use atmospheric modeling to generate atmospheric phase delays for corrections. Alternatively, time-series-based analysis, such as small baseline-subset analysis, would also remove temporally random atmospheric effects. However, this will be primarily
retrospective after a sufficient winter-time series has been captured and relies on enough temporal coherence between image pairs. Regardless, the atmospheric effects must be considered and corrected for in any practical application of InSAR SWE retrievals. Future work should include interval board measurements over a wide elevation range to identify the correct phase gradient for the InSAR retrieval.

Future work should also focus on how to best use these phase-based SWE change measurements to supplement and expand
existing snow monitoring techniques in periods and regions where the technique is most appropriate. These include accumulation periods when non-maritime snowpack tends to be drier and in higher alpine regions with fewer trees and drier snow.

Spatially, practitioners could continue to use pre-existing SNOTEL and snow-survey interpolation techniques below treeline where they are more appropriate due to less spatial variability and more uniform elevation gradients in SWE accumulation. In the complex alpine environment above treeline and in regions with less vegetation, practitioners could apply InSAR retrievals to improve the pre-existing models and capture variability and changes in SWE storage. Temporally, this technique will improve monitoring efforts in the accumulation period when current models struggle to appropriately characterize the patterns and magnitudes of precipitation events, especially in complex terrain with large amounts of wind redistribution. These InSAR phase changes will help constrain those SWE accumulation patterns before taking a less weighted effect later in the season when SWE melt-off is generally well defined by snow models. Integrating InSAR SWE retrievals into existing water monitoring systems will only improve results when appropriately applied, considering the strengths and weaknesses of the technique. Practitioners, considering the limitations mentioned above, could consider InSAR retrievals to be essentially spatially expansive, relevant, remotely-sensed SNOTEL sites.

The locations and setup for the next generation of in situ monitoring stations should also consider the needs and potential of InSAR-based SWE retrievals. Currently, snow monitoring relies on a few heavily instrumented stations to characterize entire basins. Future research should evaluate the utilization of higher densities of simply instrumented in situ stations, capturing only temperature and snow depth, across a smaller region to characterize and validate a small patch of remotely sensed data and modeling results. This smaller region would need to be carefully selected within each study basin to cover the dynamic range of controlling factors including incidence angle, aspect, elevation, and vegetation characteristics. Water forecasters could then rely on that well-validated and calibrated remotely sensed data and model output to forecast across larger regions. Specifically for InSAR phase validation, this could involve capturing SWE change over a range of elevations and aspects to validate atmospheric corrections, confirm the spatial patterns in the InSAR imagery, and remove or correct those affected by low coherence, atmospheric effects, and other biases. Future in situ monitoring sites should consider how their placements and instrumentation complements remote sensing techniques and allows for synergistic combinations with the new tools available to practitioners.

Future research should also evaluate how to use convert intuition based manual checks into automated systems for validating InSAR phase retrievals and ensuring believable spatial trends and magnitude. Most practitioners will have an intuition about likely spatial trends in precipitation that they can initially use to identify image pairs or sub-regions biased by low coherence, unwrapping errors, or biases from snow-wetness within the image. Successful application of this technique will involve finding ways to convert that intuition-based error checking and masking of regions or image pairs into automated checks of the InSAR phase that will control for images with unrealistic results to limit the biasing effects of these errors in any future processing pipelines.

Our study location, in the central mountains of Idaho, has a transitional snow climate with more mid-winter rain and SWE melt to the southwest, and a colder, often deeper snowpack to the northeast, vegetation patterns that are highly aspect and elevation-dependent and range from high-density evergreen to sagebrush to alpine treeless regions. Future work should explore this technique in maritime and continental snowpacks and other vegetation classes, such as tundra or agricultural regions. The

SnowEx data provides an opportunity to test this approach in a range of snow climates, and our future work will expand this analysis across the entire SnowEx domain.

A specific challenge of InSAR SWE retrievals is that in regions with large spatial variability unwrapped phase will be necessary to capture trends across large areas. If wrapped phase is used there may be regions that "wrap" to unreasonably low or high changes. However, phase unwrapping is computationally expensive, can introduce errors in regions of low coherence, and at low frequencies the scene-wide change variability may be effectively captured by the wrapped phase. Future work should investigate how often we will need to unwrap the InSAR phase to effectively capture snow accumulations and the best methods for unwrapping in snow covered regions.

We noted significant adverse effects on the correlation between our results and SnowModel SWE changes for periods and regions with wet snow effects. We did not attempt to parameterize or correct, potentially using modeled or elevation-based snow wetness, for these impacts. Future work on retrieval improvements that characterize and account for snow-wetness impacts will be essential to expand this technique into lower-elevations and ablation periods, which will become increasingly important with the more frequently observed mid-winter rain and melt events at greater ranges of elevations.

Finally, a range of other frequencies and SAR techniques exist that could be combined with this technique to improve retrievals and supplement the weaknesses of this technique. Future work should explore snow-depth-based surface topography approaches (SfM, lidar) in periods of wet snow, higher frequency backscatter-based approaches, and snow properties' relationship to coherence and polarimetry.

## 7   Conclusions

We present a novel comparison of a full winter-season time series of L-band InSAR SWE retrievals against in situ and modeled validation data sets. Our technique shows promise, capturing trends and absolute values of new snow accumulation. We show well matched snow depth and SWE time series from the UAVSAR retrieved values when compared against three SNOTEL stations. Comparisons between in situ observations, captured during the NASA 2020 and 2021 SnowEx campaigns, and UAVSAR retrievals showed high correlation and low RMSEs for snow depth changes (RMSE: <0.1 m and r: 0.80) and SWE change (<0.04 m and r: 0.52). The UAVSAR images also captured orographic trends in new SWE accumulations well when compared to an interval board network setup across a large elevation range for multiple storms. Finally, comparison to SnowModel SWE changes also suggest the L-band InSAR was realistically capturing SWE accumulation with strong correlation between the two (RMSD <0.023 m and r: 0.60) and especially good relationship in regions with limited snow melt and lower vegetation percentages (RMSD <0.013 m and r: 0.72). Using this SnowModel to UAVSAR comparison this study explored the controlling factors of this technique's accuracy, including the importance of appropriately applying this technique in regions of relatively-dry snow and lower vegetation percentages based on a comparison to SnowModel SWE change and SWE melt. Overall this study demonstrates the promise of using future L-band InSAR missions for snow water storage monitoring across large regions and time periods.

*Code availability.* Research code for this study is available at: https://github.com/ZachHoppinen/uavsar-validation.

*Author contributions.* Conceptualization: SO, ZH, HP. Writing: ZH, RM Analysis: ZH. Model Generation: RM. Funding Acquisition: HPM,
CV. Planning: HPM, CV, KE. Editing: ZH, HPM, KE, CV.

*Competing interests.* CV is a part of the editoral board of The Cyrosphere. The authors have no other competitions of interest.

*Acknowledgements.* This work was completed with funding from CRREL grant #W913E520C0017 and the NASA Terrestrial Hydrology
Program.
        We wish two thank our two referred reviewers (Dr. Andrea Manconi and Dr. Mathieu Le Breton) and one public reviewer (Dr. Jorge Ruiz)
for their time and thoughtful comments.

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

**Appendix A**

Figures showing the coherence (Figure **??**) and wrapped phase (Figure **??**) for the nine images used are presented below.

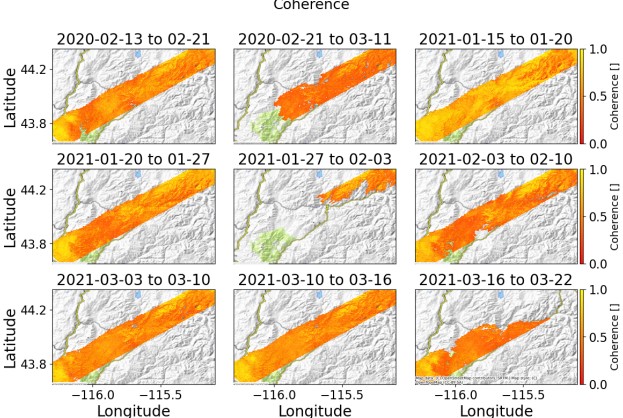

**Figure 16.** Coherence for the nine images used. Note that consistent color bounds between zero and one are used for all subplots and that the images were masked to successfully unwrapped regions.

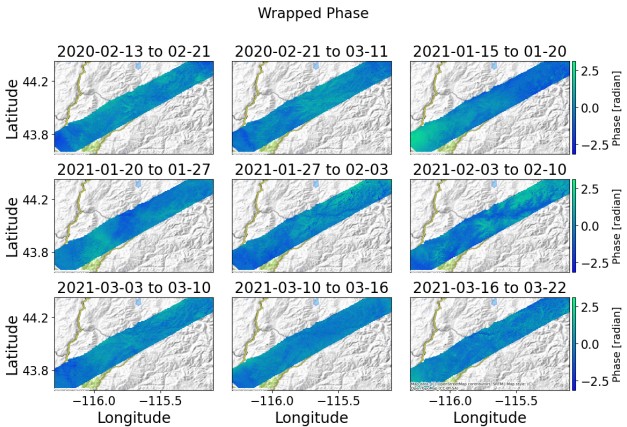

**Figure 17.** Wrapped phases for the nine images used. Note that consistent color bounds between negative pi and positive pi are used for all subplots with all means normalized to zero.