# Peer review of "Snow Water Equivalent Retrieval Over Idaho, Part B: Using L-band UAVSAR Repeat-Pass Interferometry"

_The Cryosphere, 2023_

## Author Comment (AC1)

**Review Response A** - Snow Water Equivalent Retrieval Over Idaho, Part B: Using L-band UAVSAR Repeat-Pass Interferometry**

**October 2023**

We wish to thank Dr. Andrea Manconi for her time and consideration of our study. We generally agree with her comments and provide specific replies to her comments (italicized).

*(1) The authors state several times that they "utilized wrapped images when complete spatial or temporal coverage was necessary". However, this requires a clarification, especially to readers not aware of (or not used to) the differences between wrapped and unwrapped phase in radar interferometry. I suggest providing specific details what does it mean exactly and how you combined the results of wrapped phase and unwrapped phase*

We agree that the difference between wrapped and unwrapped phase is an important and often challenging concept for readers to understand. We endeavor to provide a clear description of the wrapped vs unwrapped phase in lines 44-49. On re-reviewing that section I think that adding additional citations from (1) and (2) to provide readers in need of additional information on phase unwrapping will be helpful.

*(2) The section 3.2. is unclear. I think I get the sense of what you mean when you need for a reference phase, but the process of how you get is not straightforward (at least not in your explanation) . I suggest to write down the formulas and also add a figure showing how the reference phase looks like.*

We propose adding an equation that shows how mean phase of each scene is set using:

$$\phi_{scene}(t) = \frac{\triangle d_{insitu}(t) \times \lambda}{(4\pi)} \times \frac{1}{\cos\alpha - \sqrt{\epsilon_s(\rho_s) - sin^2\alpha}} \tag{1}$$

with $\triangle d$ representing the average change in snow depth across the in situ stations and $\epsilon$ calculated from the average density across the in situ stations.

*(a) Spatial limitation: it is true that if the InSAR retrieved deformation field is smooth and continuous, implying also appropriate spatial sampling (pixel resolution), the wrapping limit is at 2pi. However, some discontinuities in the InSAR results might occur, i.e., the phase unwrapping (which is a gradient based approach, and needs thus continuity) would fail in providing accurate results. I don't have experience with L-Band interferograms related to snow height change, thus it is difficult for me to understand if the continuity condition is respected, especially in locations with high topographic relief. Including one or more interferograms (wrapped) either in the main text or in the supplementary would help in better understanding.*

We agree that including a supplementary section showing figures with 3x3 subplots for unwrapped phase, 3x4 wrapped phase, and 3x4 for coherence so that readers could visualize the wrapped, unwrapped, and coherence images would be helpful.

*(b) Temporal limitation. The theoretical limit of phase aliasing between 2 acquisitions is = lambda/(4\*dt). With lambda L-Band ca 24 cm this means that in case of changes larger than 1.5 cm/day on the same pixel, we would reach the ambiguity limit. If the spatial unwrapping works well (see point before) then it should be not a problem. However, what happens in the cases when the phase unwrapping does not work and you use the wrapped phase values?*

First, since we are setting the reference phase it is about relative changes

so if there is a large snowfall event with less than 24 cm of difference in snow fall across the scene then phase wrapping shouldn't be an issue and we should be able to use wrapped phase. We believe in areas where we have greater than 24 centimeters of change that using the wrapped phase would be a mistake and would lead to fairly obvious 24 cm errors in regions with wrapped phase and would then need to be unwrapped using a different method (SNAPHU) or other interpolation or corrections might be necessary. We think to better clarify both the spatial and temporal limitations of this phase unwrapping processes an additional paragraph should be added to section 7.3 (Limitations) to address the need to use the unwrapped phase for large temporal baselines with over 24 cm of variation in snow depth across a scene.

*(4) related to the previous point, I find figure 7 of difficult reading. I know that it is convenient to put on a single graph several variables, but i think that for a better understanding you can put several graphs for different densities (using upper and lower boundaries) and/or different incidence angles. As mentioned in point (3a and 3b) spatial and temporal resolution play also an important role in the definition of the phase aliasing.*

I can add a second 2 axis row of plots showing additional slices through the cube to showcase two sets of densities (a lower @ 200 and an upper @ 400) along with two incidence angles (30, 60) to improve the visualization on this plot.

*(5) Missing units on the Figure 9 (y-axis)*

Thank you for catching this. We will add the appropriate units to Figure 9's y-axis.

**1 References**

**References**

[1] Rosen, P. A. *et al.* Synthetic aperture radar interferometry. *Proceedings of the IEEE* **88**, 333–382 (2000).

[2] Goldstein, R. M., Zebker, H. A. & Werner, C. L. Satellite radar interferometry: Two-dimensional phase unwrapping. *Radio Science* **23**, 713–720 (1988).

---

## Author Comment (AC2)

**Review Response B - Snow Water Equivalent Retrieval Over Idaho, Part B: Using L-band UAVSAR Repeat-Pass Interferometry**

**October 2023**

We wish to thank Dr. Mathieu Le Breton for his comments and feedback on this study. We generally agree and provide specific replies to his comments (italized) below:

*However, I am concerned by a potential methodological flaw. The article claims to retrieve the Snow Water Equivalent using aerial SAR data. Yet, on line 171-173, you say that for each image pair, you use the mean density (equivalent to dielectric permittivity) from insitu observation, in order to estimate snow depth and SWE from UAVSAR. As it is, I would rather say that the method estimates snow depth, by combining UAVSAR+insitu density measurements. SWE is then derived using again this average density.*

While we do utilize the aggregated in situ measurements to provide an estimate of insitu density it is an extremely limited contributor to the overall performance of the retrievals. InSAR measurements are quite insensitive to errors in the estimation of permittivity from dry snow density as discussed in this paper [lines 59-62], and (1) which compares extreme values of snow density and find under a 7 % error from those extreme values of snow density. Additionally (1), (2), (3) all discuss approximations for SWE directly from phase and incidence angles due to the limited impacts on SWE retrievals from snow density. Finally, (4; 5) evaluated paired snow depth and SWE measurements and found

the majority of variability derived from snow depth rather than density variations. The limited effect ($<7\%$) on performance and densities limited variability are why we say we are capturing SWE even though we do utilize the aggregated snow densities.

We do agree it is very important to clarify that this snow depth retrieval technique requires either an estimate of snow density or an approximation such as those in (1; 2; 3) in the discussion and to clarify that based on previous work our retrievals are quite insensitive to errors in this estimate of density. We think adding a few sentences to section 1.2 (Previous Work) discussing the relative variations in density vs depth (6; 5) along with the previous sensitivity analysis on density relative to phase (1) and previous SWE approximations will be a useful addition to clarify this methodological choice and can re-reference that previous work section in section 3.2 (Setting the UAVSAR Reference Phase) when we discuss using the insitu densities.

*Following this point of view, comparing results that use insitu data, with the same insitu data, seems sloppy. In consequence, accuracy estimation (figure 10, figure 11, section 7.1), one of the article's question, seems sloppy.*

Figure 11 - uses interval boards (which were not used in the density estimations or phase reference setting) and are thus independent.

Figure 10 and section 7.1 - We agree that future work with increased numbers of data points should definitely consider using completely separate validation and calibration in situ datasets. We had 64 in situ snow depth changes, and 57 swe change measurements over two winters and 10 flight dates. Since we only used the aggregated snow depth change, density across all the insitu measurements, and the scene-wide phase for each flight [discussed in section 3.2 - lines 152-160] the agreement between in situ and retrieved swe and snow depth changes will be primarily due to phase variation across the scene matching snow

depth and swe variation across the scene.

We can add in an additional sentence to Section 7.4 (Future Work) discussing the need for future analysis to evaluate the importance and limitations of setting the reference phase and using a larger in situ dataset to evaluate the impact of different method of setting the scenes reference phase.

*175: the UAVSAR timeseries represents what?* How about - "UAVSAR snow and swe depth retrieval timeseries"

*177: retrieved mean snow depth or SWE : is it snow depth or SWE ?* This line [177] should read "retrieved mean snow depth **and** swe" not or.

*5 and 118 : What is SNOTEL, how does it measure SWE ? 97 : what are the telemetered stations ?* We can add a citation (7) for those unfamiliar with the Snotel network and text such as - "Snotel Network - a system of 900 telemered stations with snow depth, snow water equivalent, and temperature. Which measures SWE with a pressure measurement from a glycol filled bladder, measuring the weight of the snowpack."

*what are the models for SWE ?*

These are 100 m resolution SNOWMODEL outputs of SWE. The SWE model is described in section 2.4 and referenced there for additional reading.

*112: Which nine pairs did you use ?* As we mention in the text [112] we used the pairs that successfully unwrapped for most analysis. For a list of those see Table 1 which has a column showing which successfully unwrapped.

*Fig 2: Maybe indicate the buffer boundary here, to clarify relation with fig 3.*

Thank you for this suggestion. We can definitely add the buffer boundary in to clarify that relation.

*Fig 3: What do you mean by clipping the data, and buffer zone ?*

Clipping - "a small piece trimmed from something". In line [190] we discuss

that we use a 1-kilometer buffer around a line connecting the interval board locations to select UAVSAR swe changes to compare to the in situ interval board swe changes. We will add in "1-kilometer buffer around a line connecting the interval board locations" to this caption to clarify what this is showing

*Fig 4: Is it cumulative precipitations ? It looks there is no melting.*

This is SWE - the x ticks on this are the date of observation and only go through April 1st before the melt season has begun. We can increase the size of the xlabels to make it clear that this is during the accumulation period.

*Fig 4: Can you add snow depth ?*

Yes, we can definitely add snow depth.

*128: Did you use Liston's model ? (you state it 'can' be used, not that you used it)*

Yes, SnowModel (developed by Liston and others) is the snow-evolution model that we used at all other points where we discuss modeled SWE. To clarify that we use this Liston model (called SnowModel) we will change "can" to "did" use and add a parenthetical reference to SnowModel in the first sentence of the dataset section where we generally describe that we used a SWE snowmodel and include the section number (2.4) where we describe SnowModel.

*132: Not clear what you used for computing the snow model. (ok: explained later) What is the altitude of the plane ? How do you compute the phase ? What is the emitted frequency exactly ? How do you ensure in your method that the phase is dominated by the ground reflexion, and not by the reflexion on the top of the snow ?*

Altitude of the plane can be incorporated into the section 2.2 (UAVSAR imagery). $\approx$13,700 m.

Phase is computed by taking the complex conjugate of one image multiplied by the other image (can be added into the second sentence of the first paragraph

of section 2.2).

Uavsar's sensor frequency is 1.26 GHz or 23.84 cm. We can add that into section 2.2 (UAVSAR imagery).

There is a decent amount of previous work analyzing where backscattered energy at different frequencies come froms in alpine snowpacks. We can add in a citation (8) that discusses where the majority of returned signals arise at different wavelengths and showing that almost all the backscattered signal returns from the snow-ground interface at L-band.

*166: What is a retrospective atmospheric to remove atmospheric phase ? Is it what is described in the previous paragraph ?*

Yes it is a reanalysis of atmospheric conditions. We will change this to use consistent terminology such as "reanalysis-based phase delay estimation" consistently between these two paragraph to clarify that the two paragraphs are discussing the same atmospheric reanalysis product.

*174: From the title of part 4, we expect results, yet this is still the method. Also, section 2 is called method (which it is) bu parts 3 and part 4, that are also methodological, are called differently. That is minor but a bit confusing.*

We agree that that section should be a subsection (3.5) instead of 4. We will change this.

*Fig. 7 is not straightforward to grasp and use. I may suggest a simpler graph more focused on the unwrapping limits. It could be for example a 2D graph with just a line representing 2pi phase shift, depending on the angle of incidence and on SWE variation (several lines for several fixed densities). It would be more informative.*

We can add an additional pair of subplots showing SWE vs incidence angles with phase visualized as a 2d colormap for 250kg/m3 and 350 kg/m3.

**1 References**

**References**

[1] Leinss, S., Wiesmann, A., Lemmetyinen, J. & Hajnsek, I. Snow Water Equivalent of Dry Snow Measured by Differential Interferometry. *IEEE Journal of Selected Topics in Applied Earth Observations and Remote Sensing* **8**, 3773–3790 (2015).

[2] Guneriussen, T., Hgda, K. A., Johnsen, H. & Lauknes, I. Insar for Estimation of Changes in Snow Water Equivalent of Dry Snow. *IEEE Transactions on Geoscience and Remote Sensing* **39**, 2101 (2001).

[3] Oveisgharan, S., Zinke, R., Keskinen, Z. & Marshall, H. Estimating Snow Water Equivalent Using Sentinel-1 Repeat-Pass Interferometry over idaho. *The Cryosphere Discussions* (2023).

[4] Sturm, M. & Wagner, A. M. Using repeated patterns in snow distribution modeling: An Arctic example. *Water Resources Research* **46** (2010).

[5] Zhao, W. *et al.* Spatial and temporal variability in snow density across the Northern Hemisphere. *CATENA* **232**, 107445 (2023).

[6] Sturm, M. *et al.* Estimating Snow Water Equivalent Using Snow Depth Data and Climate Classes. *Journal of Hydrometeorology* **11**, 1380–1394 (2010).

[7] Schaefer, G. L. & Paetzold, R. F. SNOTEL (SNOwpack TELemetry) And SCAN (Soil Climate Analysis Network). *Automated Weather Stations for Applications in Agriculture and Water Resources Management: Current Use and Future Perspectives* (2000).

[8] Naderpour, R., Schwank, M., Houtz, D., Werner, C. & Mätzler, C. Wideband Backscattering From Alpine Snow Cover: A Full-Season Study. *IEEE Transactions on Geoscience and Remote Sensing* **60**, 1–15 (2022).

---

## Author Response (AR1)

**Author Revisions and Responses - Snow Water Equivalent Retrieval Over Idaho, Part B: Using L-band UAVSAR Repeat-Pass Interferometry**

**November 2023**

We wish to thank all reviewers and public commenters for their time and consideration of our study. We present each referred and public comment below in italics, our reviewer response below that, and then an itemized list with line numbers for the revised manuscript [] highlighting change to the manuscript. Note that per a comment we have changed section 4 to section 3.5 leading to some ambiguity on section numbers. We give the revised section number along with section heading when referencing in our changes. A separate PDF highlighting these changes is provided separately.

**Reviewer A - Dr. Andrea Manconi**

*(1) The authors state several times that they "utilized wrapped images when complete spatial or temporal coverage was necessary". However, this requires a clarification, especially to readers not aware of (or not used to) the differences between wrapped and unwrapped phase in radar interferometry. I suggest providing specific details what does it mean exactly and how you combined the results of wrapped phase and unwrapped phase*

**Author Response**

We agree that the difference between wrapped and unwrapped phase is an important and often challenging concept for readers to understand. We endeavor

to provide a clear description of the wrapped vs unwrapped phase in lines 44-49. On re-reviewing that section I think that adding additional citations from (Rosen et al., 2000) and Goldstein et al. (1988) to provide readers in need of additional information on phase unwrapping will be helpful.

**Changes**

- We added two new bibliography entries for Rosen et al. (2000) and Goldstein et al. (1988) to the bibliography.

- We parenthetically included these citations in our description of phase wrapping and unwrapping [46-47]

- referred readers to section 1.1 (SAR Overview) in methods section discussing wrapped vs unwrapped images [112]

*(2) The section 3.2. is unclear. I think I get the sense of what you mean when you need for a reference phase, but the process of how you get is not straightforward (at least not in your explanation) . I suggest to write down the formulas and also add a figure showing how the reference phase looks like.*

**Author Response**

We propose adding an equation that shows how mean phase of each scene is set using:

$$\phi_{scene}(t) = \frac{\triangle d_{insitu}(t) \times \lambda}{(4\pi)} \times \frac{1}{\cos\alpha - \sqrt{\epsilon_s(\rho_s) - sin^2\alpha}} \tag{1}$$

with $\triangle d$ representing the average change in snow depth across the in situ stations and $\epsilon$ calculated from the average density across the in situ stations.

**Changes**

- We added the above equation as Equation 3 in the manuscript.

- We added some clarification of the symbols used in the equation along with a reference to the equation in the text [153-166]

- We also added a reference to Leinss et al. (2015) for further reading on swe and snow depth related phase references.

*(a) Spatial limitation: it is true that if the InSAR retrieved deformation field is smooth and continuous, implying also appropriate spatial sampling (pixel resolution), the wrapping limit is at 2pi. However, some discontinuities in the InSAR results might occur, i.e., the phase unwrapping (which is a gradient based approach, and needs thus continuity) would fail in providing accurate results. I don't have experience with L-Band interferograms related to snow height change, thus it is difficult for me to understand if the continuity condition is respected, especially in locations with high topographic relief. Including one or more interferograms (wrapped) either in the main text or in the supplementary would help in better understanding.*

**Author Response**

We agree that including a supplementary section showing figures with the wrapped phase and coherence so that readers could visualize the wrapped and coherence images would be helpful.

**Changes**

- We added an appendix with two figures showing the coherence and wrapped phase for the 9 InSAR images used.

*(b) Temporal limitation. The theoretical limit of phase aliasing between 2 acquisitions is = lambda/(4*dt). With lambda L-Band ca 24 cm this means that in case of changes larger than 1.5 cm/day on the same pixel, we would reach the ambiguity limit. If the spatial unwrapping works well (see point before) then it should be not a problem. However, what happens in the cases when the phase*

*unwrapping does not work and you use the wrapped phase values?*

**Author Response**

First, since we are setting the reference phase it is about relative changes so if there is a large snowfall event with less than 24 cm of difference in snow fall across the scene then phase wrapping shouldn't be an issue and we should be able to use wrapped phase. We believe in areas where we have greater than 24 centimeters of change that using the wrapped phase would be a mistake and would lead to fairly obvious 24 cm errors in regions with wrapped phase and would then need to be unwrapped using a different method (SNAPHU) or other interpolation or corrections might be necessary. We think to better clarify both the spatial and temporal limitations of this phase unwrapping processes an additional paragraph should be added to section 6.4 (Future work) to address the need to use the better understand when we will need to use unwrapped phase for large temporal baselines with over 24 cm of variation in snow depth across a scene.

**Changes**

- We edited section 6.4 - Future Work to include a discussion of the pros and cons of using wrapped vs unwrapped phase information for snow retrievals and the need for future work to explore when and where we will need to use both. [310, 364-369]

*(4) related to the previous point, I find figure 7 of difficult reading. I know that it is convenient to put on a single graph several variables, but i think that for a better understanding you can put several graphs for different densities (using upper and lower boundaries) and/or different incidence angles. As mentioned in point (3a and 3b) spatial and temporal resolution play also an important role in the definition of the phase aliasing.*

**Author Response**

I can add an additional subplot an showing additional slice through the cube at 30 degrees incidence angles to improve the visualization on this plot.

**Changes**

- We have added a new 2d subplot showing phases at 30 degree incidence angle along with the pre-exisiting 2d slices at 30 cm hn and 200 kg/m3.

*(5) Missing units on the Figure 9 (y-axis)*

**Author Response**

Thank you for catching this. We will add the appropriate units to Figure 9's y-axis.

**Changes**

- Added "Depth [m]" to the y-axis of subplots on Figure 9.

**Reviewer B - Dr. Mathieu Le Breton**

*However, I am concerned by a potential methodological flaw. The article claims to retrieve the Snow Water Equivalent using aerial SAR data. Yet, on line 171-173, you say that for each image pair, you use the mean density (equivalent to dielectric permittivity) from insitu observation, in order to estimate snow depth and SWE from UAVSAR. As it is, I would rather say that the method estimates snow depth, by combining UAVSAR+insitu density measurements. SWE is then derived using again this average density.*

**Author Response**

While we do utilize the aggregated in situ measurements to provide an estimate of insitu density it is an extremely limited contributor to the overall performance of the retrievals. InSAR measurements are quite insensitive to errors in the estimation of permittivity from dry snow density as discussed in this paper [lines 59-62], and (Leinss et al., 2015) which compares extreme values of

snow density and find under a 7 % error from those extreme values of snow density. Additionally (Leinss et al., 2015), (Guneriussen et al., 2001), (Oveisgharan et al., 2023) all discuss approximations for SWE directly from phase and incidence angles due to the limited impacts on SWE retrievals from snow density. Finally, (Sturm and Wagner, 2010; Zhao et al., 2023) evaluated paired snow depth and SWE measurements and found the majority of variability derived from snow depth rather than density variations. The limited effect ($<7\%$) on performance and densities limited variability are why we say we are capturing SWE even though we do utilize the aggregated snow densities.

We do agree it is very important to clarify that this snow depth retrieval technique requires either an estimate of snow density or an approximation such as those in (Leinss et al., 2015; Guneriussen et al., 2001; Oveisgharan et al., 2023) in the discussion and to clarify that based on previous work our retrievals are quite insensitive to errors in this estimate of density. We think adding a few sentences to section 1.1 (SAR Overview) discussing the relative variations in density vs depth (Sturm et al., 2010; Zhao et al., 2023) along with the previous sensitivity analysis on density relative to phase (Leinss et al., 2015) and previous SWE approximations will be a useful addition to clarify this methodological choice and can re-reference that previous work section in section 3.2 (Setting the UAVSAR Reference Phase) when we discuss using the insitu densities.

**Changes**

- We added a paragraph to section 1.1 - SAR Overview to highlight the limited impact of density estimates on snow depth retrievals from phase using Leinss et al. (2015) and to cite other studies that use an approximation to get SWE from phase (Leinss et al., 2015; Guneriussen et al., 2001; Oveisgharan et al., 2023). [64-66]

- We added Leinss et al. (2015); Guneriussen et al. (2001); Oveisgharan

et al. (2023) to the references

- We also added a paragraph to section 6.3 - Limitations discussing that since we did not use a SWE approximation this study's methods require some estimate of density to perform [302-306]

*Following this point of view, comparing results that use insitu data, with the same insitu data, seems sloppy. In consequence, accuracy estimation (figure 10, figure 11, section 7.1), one of the article's question, seems sloppy.*

**Author Response**

Figure 11 - uses interval boards (which were not used in the density estimations or phase reference setting) and are thus independent.

Figure 10 and section 7.1 - We agree that future work with increased numbers of data points should definitely consider using completely separate validation and calibration in situ datasets. We had 64 in situ snow depth changes, and 57 swe change measurements over two winters and 10 flight dates. Since we only used the aggregated snow depth change, density across all the insitu measurements, and the scene-wide phase for each flight [discussed in section 3.2 - lines 152-160] the agreement between in situ and retrieved swe and snow depth changes will be primarily due to phase variation across the scene matching snow depth and swe variation across the scene.

We can add in an additional sentence to Section 7.4 (Future Work) discussing the need for future analysis to evaluate the importance and limitations of setting the reference phase and using a larger in situ dataset to evaluate the impact of different method of setting the scenes reference phase.

**Changes**

- We added in additional language to section 6.3 - Limitations where we directly discuss the fact that we chose to use aggregated in situ measurements to set the scene-wide mean phase due to our limited number of in situ observations. [307-314]

*175: the UAVSAR timeseries represents what?*

**Author Response**

We will clarify this language to show that we are comparing three SNOTEL SWE and snow depth measurements against UAVSAR retrievals of SWE and snow depth

**Changes**

- We have changed these lines to better clarify that we are comparing a time series of snow and SWE retrievals from SNOTEL against UAVSAR. [184-185]

*177: retrieved mean snow depth or SWE : is it snow depth or SWE ?*

**Author Response**

This line [177] should read "retrieved mean snow depth **and** swe" not or.

**Changes**

- Changed this from "or" to "and"

*5 and 118 : What is SNOTEL, how does it measure SWE ? 97 : what are the telemetered stations ?*

**Author Response**

We can add a citation (Schaefer and Paetzold, 2000) for those unfamiliar with the Snotel network and text such as - "Snotel Network - a system of 900 telemered stations with snow depth, snow water equivalent, and temperature. Which measures SWE with a pressure measurement from a glycol filled bladder, measuring the weight of the snowpack."

**Changes**

- We added Schaefer and Paetzold (2000) to the references

- We have added a parenthetical reference to Schaefer and Paetzold (2000) for those unfamiliar with SNOTEL [123].

- We also added two sentence explaining more about snotel and how it measures SWE [125-127].

*what are the models for SWE ?*

**Author Response**

These are 100 m resolution SNOWMODEL outputs of SWE. The SWE model is described in section 2.4 and referenced there for additional reading.

**Changes**

*112: Which nine pairs did you use ?*

**Author Response**

As we mention in the text [112] we used the pairs that successfully unwrapped for most analysis. For a list of those see Table 1 which has a column showing which successfully unwrapped.

**Changes**

- We clarify the language in Section 2.2 - UAVSAR imagery to clarify that we use the 9 images that succesfully unwrapped for most of our analysis and reference table 1 to clarify which images were used [115-117]

*Fig 2: Maybe indicate the buffer boundary here, to clarify relation with fig 3.*

**Author Response**

Thank you for this suggestion. We can definitely add the buffer boundary in to clarify that relation.

**Changes**

- We added the buffer boundary to figure 2 and edited the legend and caption to match.

*Fig 3: What do you mean by clipping the data, and buffer zone ?*

**Author Response**

Clipping - "a small piece trimmed from something". In line [190] we discuss that we use a 1-kilometer buffer around a line connecting the interval board locations to select UAVSAR swe changes to compare to the in situ interval board swe changes. We will add in "1-kilometer buffer around a line connecting the interval board locations" to this caption to clarify what this is showing

**Changes**

- Changed Figure 3's caption to: "Map of the interval board locations and one-kilometer buffer around a line connecting the interval board locations used to clip the UAVSAR data"

*Fig 4: Is it cumulative precipitations ? It looks there is no melting.*

**Author Response**

This is SWE - the x ticks on this are the date of observation and only go through April 1st before the melt season has begun.

**Changes**

- We changed this y label to "SNOTEL Depth" to hopefully make it clear that this is not cumulative precipitation

*Fig 4: Can you add snow depth ?*

**Author Response**

Yes, we can definitely add snow depth.

**Changes**

- We have added snow depth to figure 4 and adjusted the y label, legend and caption appropriately.

*128: Did you use Liston's model ? (you state it 'can' be used, not that you used it)*

**Author Response**

Yes, SnowModel (developed by Liston and others) is the snow-evolution model that we used at all other points where we discuss modeled SWE. To clarify that we use this Liston model (called SnowModel) we will change "can" to "did" use and add a parenthetical reference to SnowModel in the first sentence of the dataset section where we generally describe that we used a SWE snowmodel and include the section number (2.4) where we describe SnowModel.

**Changes**

- Changed "can be" to "was" [133]

- Added parenthetical reference to Liston et al.'s model when we first introduce modeled data sets in section 2.1 (Liston and Elder, 2006; Liston et al., 2020)

*132: Not clear what you used for computing the snow model. (ok: explained later) What is the altitude of the plane ? How do you compute the phase ? What is the emitted frequency exactly ? How do you ensure in your method that the phase is dominated by the ground reflexion, and not by the reflexion on the top of the snow ?*

**Author Response**

Altitude of the plane can be incorporated into the section 2.2 (UAVSAR imagery). ≈13,700 m.

Phase is computed by taking the complex conjugate of one image multiplied by the other image (can be added into the second sentence of the first paragraph of section 2.2).

Uavsar's sensor frequency is 1.26 GHz or 23.84 cm. We can add that into section 2.2 (UAVSAR imagery).

There is a decent amount of previous work analyzing where backscattered energy at different frequencies come froms in alpine snowpacks. We can add in a citation (Naderpour et al., 2022) that discusses where the majority of returned signals arise at different wavelengths and showing that almost all the backscattered signal returns from the snow-ground interface at L-band.

**Changes**

- We added the plane height and sensor frequency to section 2.2 [107]

- We add a sentence explaining how InSAR phase is calculated (in addition to it being explained in both citations) [113-114]

- Added a citation - Naderpour et al. (2022) to line describing how most of the energy is refracted from the snow-ground interface for low frequency radars [39]

*166: What is a retrospective atmospheric to remove atmospheric phase ? Is it what is described in the previous paragraph ?*

**Author Response**

Yes it is a reanalysis of atmospheric conditions. We will change this to use consistent terminology such as "reanalysis-based phase delay estimation" consistently between these two paragraph to clarify that the two paragraphs are discussing the same atmospheric reanalysis product.

**Changes**

- Changed retrospective to ERA5 reanalysis to clarify it is the same as what is discussed in the preceding paragraph [180]

*174: From the title of part 4, we expect results, yet this is still the method. Also, section 2 is called method (which it is) bu parts 3 and part 4, that are also methodological, are called differently. That is minor but a bit confusing.*

**Author Response**

We agree that that section should be a subsection (3.5) instead of 4. We will change this.

**Changes**

- Changed from a section 4 to subsection 3.5

*Fig. 7 is not straightforward to grasp and use. I may suggest a simpler graph more focused on the unwrapping limits. It could be for example a 2D graph with just a line representing 2pi phase shift, depending on the angle of incidence and on SWE variation (several lines for several fixed densities). It would be more informative.*

**Author Response**

We can add an additional pair of subplots showing SWE vs incidence angles with phase visualized as a 2d colormap for 30 degrees of incidence angle and a plot of phase unwrapping limits.

**Changes**

- We have added a new 2d subplot showing phases at 30 degree incidence angle along with the pre-exisiting 2d slices at 30 cm hn and 200 kg/m3.

- We added another subplot highlighting the phase unwrapping limits for varying incidence angles and new snow amounts at 100kg/m3 and 250kg/m3 to show the limits of wrapped phase.

**Public Comments - Dr. Jorge Ruiz**

*I am not completely sure about Section 5. I think it overlooks a bit the fact that when snow is wet the imaginary part of the complex permittivity increases. In the method you use in the article dry snow is assumed as a precondition. When snow is wet, there are losses in the medium, and the main backscatter interface changes from the ground-snow to the air-snow or the snow volume*

*(depending on the frequency and the amount of liquid water). Moreover, drops in coherence are linked to snow wetness. Could comment a bit on this? Some literature: Section IV.A from [1], Section III.A from [2], Section III in [3].*

We agree that wet snow is a significant challenge for InSAR snow depth and SWE retrievals. The UAVSAR's flights were intentionally planned to all occur during the accumulation season when the majority of snow in this region of the United States would be dry hopefully limiting the effects you are pointing to. At some of the lower elevations we do suspect that wet snow affected our retrievals and that is why we included the figures showing the potential errors in our results caused by newly wet snow [figure 8] and we discuss the likely negative impacts of this wet snow in lines [280-284]. However we agree that including [1, 4] into the initial discussions of wet snow's impacts and clarifying that it will cause not only unaccounted for changes in wave speed but will also decrease coherence [4] and the phase centroid [1] will be valuable for those considering using this technique in regions with large amounts of wet snow.

- We included Naderpour et al. (2022) and Ruiz et al. (2022)

*Was any of the retrievals affected by wet snow? I am thinking of Figure 6, pair 2021-03-03 to 03-10 and onwards. These seem to observe a decrease in SWE between passes but in Figure 4 SWE looks relatively steady... How was the coherence? Could it be rain on snow?*

As we mention in the text there was probably significant effects from wet snow that we attempt to identify and highlight in figure 13 and 15 and in our discussion of factors affecting the accuracy of retrievals.

*Figure 4: It is hard to say, but is perhaps a purple line missing, corresponding to 2021-03-22?*

Figure 4 - as per another comment we will be increasing the font size of the x-ticks to help clarify what dates are presented in this figure.

- increased font size for x-ticks

*Line 75: there is a more complete journal article [4].*

Line 75 - thank you. We will update this citation.

- Updated.

*Line 182: how many pixels or looks is the 100m box?*

line 182 - approximately 20 looks. We can parenthetically include that in this line as well to clarify.

- added parenthetical 20 looks to this line

**1 References**

**References**

Goldstein, R. M., Zebker, H. A., and Werner, C. L.: Satellite radar interferometry: Two-dimensional phase unwrapping, Radio Science, 23, 713–720, https://doi.org/10.1029/rs023i004p00713, 1988.

Guneriussen, T., Hgda, K. A., Johnsen, H., and Lauknes, I.: Insar for Estimation of Changes in Snow Water Equivalent of Dry Snow, IEEE Transactions on Geoscience and Remote Sensing, 39, 2101, https://doi.org/10.1109/36.957273, 2001.

Leinss, S., Wiesmann, A., Lemmetyinen, J., and Hajnsek, I.: Snow Water Equivalent of Dry Snow Measured by Differential Interferometry, IEEE Journal of Selected Topics in Applied Earth Observations and Remote Sensing, 8, 3773–3790, https://doi.org/10.1109/jstars.2015.2432031, 2015.

Liston, G., Stroeve, J., Buzzard, S., Zhou, L., Mallett, R., Barrett, A., Tschudi, M., Tsamados, D. M., Itkin, P., and Stewart, J.: A Lagrangian Snow-Evolution System for Sea Ice Applications (SnowModel-LG): Part II - Analyses, 125, https://doi.org/10.1029/2019jc015900, 2020.

Liston, G. E. and Elder, K.: A Distributed Snow-Evolution Modeling System (SnowModel), Journal of Hydrometeorology, 7, 1259–1276, https://doi.org/10.1175/jhm548.1, 2006.

Naderpour, R., Schwank, M., Houtz, D., Werner, C., and Mätzler, C.: Wideband Backscattering From Alpine Snow Cover: A Full-Season Study, IEEE Transactions on Geoscience and Remote Sensing, 60, 1–15, https://doi.org/10.1109/tgrs.2021.3112772, 2022.

Oveisgharan, S., Zinke, R., Keskinen, Z., and Marshall, H.: Estimating Snow Water Equivalent Using Sentinel-1 Repeat-Pass Interferometry over idaho, The Cryosphere Discussions, 2023.

Rosen, P. A., Hensley, S., Joughin, I. R., Li, F. K., Madsen, S. N., RodrÍgues, E., Goldstein, R. M., Bamler, R., and Hartl, P.: Synthetic aperture radar interferometry, Proceedings of the IEEE, 88, 333–382, https://doi.org/10.1088/0266-5611/14/4/001, 2000.

Ruiz, J. J., Lemmetyinen, J., Kontu, A., Tarvainen, R., Vehmas, R., Pulliainen, J., and Praks, J.: Investigation of Environmental Effects on Coherence Loss in SAR Interferometry for Snow Water Equivalent Retrieval, IEEE Transactions on Geoscience and Remote Sensing, 60, 1–15, https://doi.org/10.1109/tgrs.2022.3223760, 2022.

Schaefer, G. L. and Paetzold, R. F.: SNOTEL (SNOwpack TELemetry) And SCAN (Soil Climate Analysis Network), Automated Weather Stations for Applications in Agriculture and Water Resources Management: Current Use and Future Perspectives, 2000.

Sturm, M. and Wagner, A. M.: Using repeated patterns in snow distribution modeling: An Arctic example, Water Resources Research, 46, https://doi.org/10.1029/2010wr009434, 2010.

Sturm, M., Taras, B., Liston, G. E., Derksen, C., Jonas, T., and Lea, J.: Estimating Snow Water Equivalent Using Snow Depth Data and Climate Classes, Journal of Hydrometeorology, 11, 1380–1394, https://doi.org/10.1175/2010jhm1202.1, 2010.

Zhao, W., Mu, C., Han, L., Sun, W., Sun, Y., and Zhang, T.: Spatial and temporal variability in snow density across the Northern Hemisphere, CATENA, 232, 107 445, https://doi.org/10.1016/j.catena.2023.107445, 2023.